# The Role of Social Capital Facing Pandemic COVID-19 in Tourism Village to Support Sustainable Agriculture (Empirical Evidence from Two Tourism Villages in Indonesia)

Aidha Auliah, Gunawan Prayitno *, Ismu Rini Dwi Ari, Rahmawati, Lusyana Eka Wardani and Christia Meidiana

Regional and Urban Planning Department, Faculty of Engineering, University of Brawijaya, Malang 65142, Indonesia
* Correspondence: gunawan_p@ub.ac.id

**Abstract:** Pujon Kidul and Bangelan villages (Indonesia) are two tourist villages that have succeeded in developing agricultural potential associated with tourism activities. The characteristics of Pujon Kidul Village and Bangelan Village refer to the tourism activities in the villages. This study aims to find out how community social capital can influence collective action in dealing with the COVID-19 pandemic in Pujon Kidul Tourism Village and Bangelan Tourism Village. This study uses the structural equation model (SEM) method, which is applied to the SEM, PLS (Partial Least Square) and AMOS (Analysis of Moment Structures) software to get complete results. The results of the analysis, Bangelan Village and Pujon Kidul Village are models that can survive in a pandemic crisis situation. The results of the model show that social capital in dealing with a pandemic in Bangelan Village is most influenced by trust in the government, while in Pujon Kidul Village it is most influenced by trust among fellow communities. The existence of social capital plays a role in decision-making on collective actions to deal with pandemics in tourist villages. Thus, by knowing how the community is recovering during the COVID-19 pandemic, businesses can run smoothly and according to government policy.

**Keywords:** social capital; pandemic; collective action; tourism village; structural equation model

## 1. Introduction

A tourist village is the attraction of village life with its unique and distinctive characteristics (cultural and natural attractions) so that it can be used as a tourist attraction that can blend in with the community and their lives (Purwanggono 2016). The experience gained from the tourism village is the unique potential of life and the traditions of rural communities (Kementerian Pariwisata 2019). A tourism village is a form of community-based tourism development that is able to develop a rural and sustainable economy (Arida et al. 2019; Nugraha et al. 2021). Tourism village development is one method of sustainable development (Scheyvens and Laeis 2021; Pasanchay and Schott 2021). Even the central and regional governments identify the development of rural tourism as one of the keys to economic development in rural areas (Davis and Morais 2004; Nugraha et al. 2021). The development of a tourist village provides both social and economic changes, such as increasing the income and welfare for rural communities (Wardani et al. 2022; Prayitno et al. 2022a). Through local rural attractions, it will be able to instill a sense of pride in the local community towards the village where they live (Aji 2020). However, with the growth of tourist activities in the village, a new challenge is introduced, namely the COVID-19 pandemic. The pandemic has caused important sectors in Indonesia, such as the tourism sector, to experience various problems (Utami 2021).

The pandemic has had both economic and social impacts on rural communities (Sie Pujonkidul Desa 2020). One form of community readiness in dealing with a pandemic is

by utilizing local wisdom. Therefore, the study of community social capital is important in strengthening the community through the COVID-19 pandemic (Prayitno et al. 2022a). The community's active role is an important point in the continuity of tourism activities in rural areas (Prayitno et al. 2022b; Soebagjo 1991). Social capital exists and is formed in society, namely from the actions of individuals developing and maintaining relationships with one another (Woolcock 1998; Light and Angeles 2001). Social capital is the utilization of resources by a group or individual from relationships that are established and bound by shared values and norms to encourage cooperation (Kawachi et al. 2008; Borgonovi and Andrieu 2020). The components of social capital as a key driver in achieving common goals. Social capital is a description of social life that plays an important role in the strengthening and functioning of the community to be actively involved in a collective action to achieve common goals (Putnam 2004; Prayitno et al. 2022c).

Researchers widely use some perspectives that include; first, that social capital occurs and takes place by providing benefits on an individual and group scale (van der Gaag 2005; Grootaert and Van Bastelar 2002). In the first perspective, many believe social capital is created by interactions between individuals within a group, which are embedded within social relations. In addition, from another perspective, social capital can facilitate collective action that aims to improve the welfare of certain groups (Lin 2001). Social capital has a major role in facilitating an act of coordination and cooperation between communities to achieve common interests (Grootaert and Van Bastelar 2002). A type of social capital can be defined by trust, networks, and the formal/informal rules of collective action (Ostrom and Ahn 2009). Likewise, in dealing with the COVID-19 pandemic, social capital can be utilized as a shared strength that facilitates collective action to unite communities in realizing a joint response to the COVID-19 pandemic crisis situation (Makridis and Wu 2021). Adopting the actions taken in China, the government should modify new policies according to the spread of the COVID-19 virus. Handling is mostly done with appropriate steps such as banning gatherings, closing schools and public places, and implementing social quarantine to limit the spread of the virus. In addition, each case must also be easily and quickly reviewed and monitored using new data and technology (Abbass et al. 2020). Indonesia itself is subject to similar policies by imposing physical restrictions and closing crowded places, such as public places, tours, etc. The tourist village has been one of the most affected by the pandemic, due to a decrease in tourist visits caused by the closure of the tourist village during the COVID-19 pandemic. Several studies have stated that crisis events in tourist destinations can have a negative impact on the emotions, attitudes, and behavior of tourists, which in turn, affect sustainable tourism development. A crisis event can have a negative impact on society. Cases of negative events in tourism can influence perceptions of betrayal and boycotts among tourists (Su et al. 2022). The existence of a crisis event in a destination can also have an impact on tourists' emotions and forgiveness, where a crisis situation that occurs externally will garner more sympathy and forgiveness than an internal crisis event (Su et al. 2022). This study, on the other hand, will go into greater detail about the COVID-19 pandemic crisis, particularly in developing tourism villages, in dealing with the COVID-19 pandemic, and in further supporting sustainable agriculture.

The use of social capital in economic recovery during the COVID-19 pandemic is implemented in economic practice in the form of collaboration between community members to produce quality production goods while maintaining prices. Villagers can take advantage of strong social networks and mutual trust to encourage creative, innovative, and productive behaviors, even during the COVID-19 pandemic crisis (Bai et al. 2020). Important social capital plays a key role in the recovery of society, both economically and socially. Social capital is able to drive economic growth and create a social cohesion that contributes to improving people's health and welfare (Makridis and Wu 2021). During the crisis conditions of the COVID-19 pandemic, social capital was able to increase community capacity to prepare for, respond to, and recover from crises (Reininger et al. 2013). Social capital in the form of trust among community members leads to the greater sharing of information about facts, procedures, or threats to the community, which is especially important

when dealing with extreme events and shocks. Social capital is important not only during a crisis, but also afterward during recovery (Makridis and Wu 2021). Social capital becomes the bond that binds the community and serves as the main capital for long-term recovery (Aldrich 2010), especially in the development of tourist villages to support sustainable agriculture. Sustainable agriculture plays an important role in supporting and balancing the development of tourist villages. By supporting sustainable agriculture, people can improve their quality of life in the agricultural sector with adequate social and economic conditions. Some farmers generally cannot protect their land due to the economic and social pressures that encourage land conversion. This causes agricultural land owners to be unable to continue farming, so that sustainable agriculture is also difficult to maintain because agricultural production has decreased (Prayitno et al. 2021). Therefore, to support sustainable agriculture in tourist villages, various forms of business are carried out by the community by involving agricultural management and maintaining and preserving what is in the present and what will be in the future, as well as by adapting to changes that occur (Nugraha et al. 2022).

Several previous studies that discussed social capital, collective action, and the COVID-19 pandemic included research (Prayitno et al. 2022b) that discussed decision making in tourist villages, based on social capital during the COVID-19 pandemic, by examining social capital in the form of beliefs, norms, and social networks connected with collective action in one village. Similarly, research (Hunecke et al. 2017) examines the role of social capital in farmer decision making regarding technology adoption, where extension efforts to adopt technology must consider social networks in promoting agricultural innovation. Subsequent research (Prayitno et al. 2022d) examined social capital and quality of life in supporting sustainable agriculture and food security. This research looks at social capital and its relation to the quality of life of rural communities, especially farmers, to support sustainable agriculture. Unlike previous research, this research raises the topic of the role of social capital in facing the COVID-19 pandemic in tourist villages to support sustainable agriculture in two tourist villages, namely Pujon Kidul Tourism Village and Bangelan Tourism Village, Indonesia. Empirical evidence from these two villages is important in order to study tourism villages with different characteristics and examine the community's recovery model in dealing with the COVID-19 pandemic from a social perspective.

Pujon Kidul Tourism Village and Bangelan Tourism Village are tourist villages located in Malang Regency, with agricultural potential that is utilized as a tourist attraction. The existence of the COVID-19 pandemic has not only reduced the number of tourist visits and the income of the people in the two villages, but has also had a social impact, due to restrictions on activities from government policies in tourism villages. Therefore, community efforts are needed to deal with the COVID-19 pandemic in tourist villages. This study aims to provide the role of community social capital as a factor for collective action related to efforts to deal with COVID-19. This research develops a structural model of community social capital in an effort to deal with the COVID-19 pandemic. The research model will be evaluated in Pujon Kidul Tourism Village and Bangelan Tourism Village. This research will determine how social capital influences collective action in dealing with the COVID-19 pandemic, which in turn, affects adaptability. Communities with good social capital will progress even in times of crisis. Thus, through a study of social capital in the two study areas, namely tourist villages with agricultural characteristics, it will provide community resilience and readiness in dealing with the COVID-19 pandemic and further promote sustainable agriculture.

## 2. Literature Review

### 2.1. Social Capital in Economic Recovery in the Face of the COVID-19 Pandemic

According to (Putnam 1993), social capital consists of trust, norms, and networks which among them constitute a single unit of capital that can be utilized in achieving common goals. Social capital forms social networks by enforcing norms that foster trust among community members, giving the community strength to overcome difficult conditions. If



community members believe that other members are trustworthy and honest, then they will trust each other (Yang and Ren 2020). The implementation of social capital through economic practices can be in the form of cooperation between community members to produce quality domestically produced goods while maintaining their prices. The target market for this production activity is the surrounding community, so that a strong network and sense of trust will be created between members of the community. This situation will encourage the community to remain creative, innovative, and productive, so that even in difficult conditions, village communities can survive the COVID-19 pandemic (Bai et al. 2020). Social capital is not only used to respond to disasters such as the COVID-19 pandemic, but is also used in preparing for preparedness, mitigation, and response and recovery so that the community can be independent and work together in dealing with disasters that occur (Bartscher et al. 2021).

As a form of recovery and strengthening of the economy during the COVID-19 pandemic, social capital plays a very important role to survive. Social capital in the economy can be used very effectively to find new positions and get promotions, and networks in social capital are seen as capital that can drive success business or activity. A network that is based on trust and limited by norms can provide an important source of information and help provide access to finance (Elgar et al. 2020). Social capital is needed in order to secure the local community's economy, especially people who develop home industries as well as the majority of people who are farmers (Pitas and Ehmer 2020). Social capital is productive, which can help economic recovery as a result of the COVID-19. As an example, in rural communities, where the majority are farmers, social capital can be utilized in lending agricultural equipment based on trust between fellow farmers. In addition, the network can be used to obtain information and markets for selling agricultural products at higher prices (Tittonell et al. 2021).

### 2.2. Collective Action in the Face of the COVID-19 Pandemic

In general, collective action is understood as the actions taken by individuals as members of a group to achieve group goals, such as social change. Collective action is also an action taken by individuals to achieve common goals (Hattke and Martin 2020). Collective action in the COVID-19 pandemic in the form of actions taken to deal with the pandemic. In the midst of the difficult situation of the COVID-19 pandemic, collective actions emerged from community and government initiatives as a form of effort being made to deal with the COVID-19 pandemic (Wilson et al. 2020). Collective actions that are often carried out include distributing free masks to the public, taking the initiative to create private handwashing stations and public handwashing areas that anyone can use, spreading information that it is important to maintain cleanliness and environmental health, inviting and implementing health protocols such as behaviors to maintain distance, washing hands properly, wearing masks, and preventing people from gathering (Harring et al. 2021). Actions from society should be taken by everyone, but a collective push is needed to ensure that everyone takes them. The idea that other people should not change their behavior encourages parties to facilitate efforts to change the expected behavior (Meinzen-Dick 2020). Therefore, the role of social capital is needed to facilitate collective action. Social capital can increase public trust in fellow citizens and trust in the government, so that it will make it easier to realize the success of collective action in the context of dealing with the COVID-19 pandemic (Wu 2021). The existence of norms, networks, and trust as elements of social capital encourages people to take collective action. These three elements function as a means of unifying goals and determinants of collective action to make them happen (Wong and Kohler 2020).

### 3. Materials and Methods

This research was conducted in Pujon Kidul Village and Bangelan Village, Indonesia. Pujon Kidul Village is one of 11 villages located in Pujon District, Malang Regency, where the entire area is included in the highlands surrounded by hills. The Pujon Kidul Village

area itself is divided into three hamlets, including Telungrejo Hamlet, Krajan Hamlet, and Maron Hamlet. Meanwhile, Bangelan Village is a village in Wonosari Regency, Malang Regency, which is divided into four hamlets, namely Bangelan Hamlet, Arjomulyo Hamlet, Sidomulyo Hamlet, and Kampung Baru Hamlet. The study was conducted by collecting data on community social capital during the COVID-19 pandemic in the two study areas.

### 3.1. Research Variable

This study examines social capital variables using the structural equation model (SEM), which allows identification of relationships between social capital variables. In identifying the role of social capital in dealing with pandemics in tourist villages, it is necessary to understand that community social capital is formed from trust, norms, and social networks. The role of social capital can be one of the efforts in dealing with the pandemic crisis, where the norms, trust, and social networks that are formed in the community allow them to more easily access various sources, such as information, assistance, and other shared resources, as a form of support between communities (Reininger et al. 2013). Social capital is broadly defined as a feature of social organization, such as social networks, norms, and beliefs that facilitate action and cooperation for mutual benefit (Putnam 1993). People with high social capital will be more active to act in the common good. the existence of social capital, in the form of ties and networks that facilitate collective action, will unite the community to realize a common handling in the face of the COVID-19 pandemic (Makridis and Wu 2021). Therefore, in this study, we try to see the role of social capital in dealing with the pandemic by modeling the relationship of community social capital in tourist villages, namely Pujon Kidul Tourism Village and Bangelan Tourism Village.

This study examines social capital with variables of trust, norms, and social networks. The trust variable consists of indicators of trust in neighbors (K1), trust in people of different ethnicities and cultures (K2), trust in village government (K3), trust in community leaders (customary leaders) (K4), trust in religious leaders (K5), trust in village institutions (K6), and trust in communication between communities (K7). Norm variables consist of indicators of adherence to customary norms/rules (N1), existence of sanctions against norms (N2), and participation in traditional activities/events (N3). The social network variable consists of indicators of community willingness to cooperate (J1), participation in religious activities/events (J2), participation in social activities/activities (J3), activeness in providing suggestions during meetings/meetings (J4), and participation in group meetings (J5).

### 3.2. Sample

This research was conducted using qualitative and quantitative approaches. The qualitative methods will be used to explain the community's characteristics and the characteristics of Pujon Kidul and Bangelan tourism villages. In contrast, the quantitative approach will describe the relationship of social capital variables. Qualitative research uses data derived from primary data following the actual situation obtained through observations and interviews. While the quantitative approach analyzes numerical data, statistical methods are used to process it (Azwar 2013). The number of samples calculated using the Krejcie–Morgan approach, with the selected degrees of freedom, is 5% or 0.5, implying that the 5% error rate calculation is acceptable. Therefore, Pujon Kidul Village has a sample of 334 people (because it has a population of 7725 people or close to 8000 people), and Bangelan Village has a sample of 323 people (because the population is 4437 people or close to 4500 people). In measuring social capital, a questionnaire was used to ask questions in accordance with social capital indicators. The data obtained are in the form of a five-point Likert scale rating from a value of 1 to 5 from strongly disagree, disagree, quite agree, agree, to strongly agree.

### 3.3. Analysis Method

This study uses structural equation model (SEM) analysis to analyze social capital in dealing with pandemics, such as COVID-19. SEM analysis is useful to connect research

variables. Several types of software used in analyzing the SEM model are AMOS (Analysis of Moment Structures) 22.00 version, LISREL(Linear Structural Relationship) 8.80 version, PLS (Partial Least Square) SmartPLS 3.0 version, and GSCA, (Generalized Structural Component Analysis). This study uses SEM-PLS and SEM-AMOS to produce models of community social capital in Pujon Kidul Village and Bangelan Village in dealing with pandemics, such as COVID-19. SEM-PLS is an analysis modeling SEM using software (Nikmatus Sholiha and Salamah 2015; Ghozali 2008). Partial least square, or PLS, is often used in research analysis methods that have a low dependence on the measurement scale. The scale measurement here is the distribution and size of the research sample (Lin 2001). AMOS is also a specialized software in SEM model analysis, with the advantage of being easy to use for beginners in the SEM field. In contrast to PLS, AMOS pays attention to large sample sizes and has reflective indicators. Several steps are taken in analyzing the SEM model, namely (1) reviewing hypotheses, the literature, and theories for the SEM model, (2) building/developing a theoretical framework in SEM, (3) developing the model specifications, (4) setting the samples, (5) calculating the research parameter estimates, (6) conducting a model feasibility test with GOF (Goodness of Fit Test) test, (7) modifying the SEM model, and (8) writing and developing a discussion of findings, suggestions, policy implications, and conclusions (Haryono 2014).

## 4. Results

### *4.1. Characteristics of Tourism Villages*

#### 4.1.1. Pujon Kidul Village

Pujon Kidul Village is engaged in the agricultural sector, which is supported by the village tourism sector. As a tourist village, Pujon Kidul Village has community activities engaged in the agricultural sector and used for tourism. Therefore, the tourist village highlights the potential of agriculture as its main attraction (Figure 1). The main attraction in Pujon Kidul Village is the Sawah Cafe, which serves a cafe tour with a view of the rice fields.

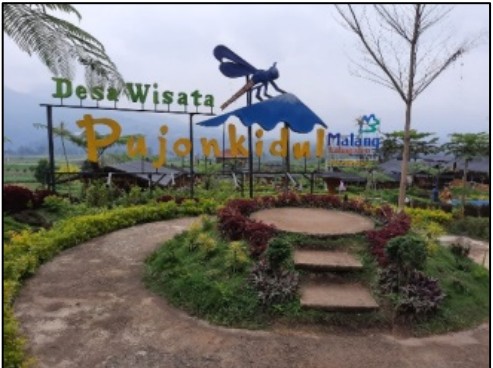 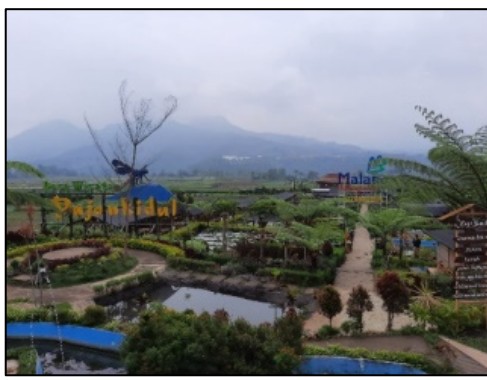

**Figure 1.** Pujon Kidul's Café Sawah. Sumber: Primary Survey Results (2021).

This tourist destination features a stretch of hills and mountains with unspoiled natural scenery. Various exciting tourist attractions that visitors can enjoy include rice field cafe areas, adventure tours, cultural park tours, agricultural education tours, fantasy land tours, and the roudh 78 tours. However, Pujon Kidul Village has experienced new shocks from the pandemic crisis, as it entered the year 2020, where the pandemic had a major impact on village tourism, such as the closure of tourism because of government policies.

#### 4.1.2. Bangelan Village

Bangelan Village, also known as the Agricultural Tourism Village, was established on 5 September 2020, by the Regent of Malang Regency. Bangelan Tourism Village focuses on developing village potential, namely the Boer goat center, beef cattle center, compost fertilizer processing center, rice food center, and the tourism waterfall, Tanaka. In 2021, the

Bangelan Village Government had a target to inaugurate tourist attractions, such Mina Padi and Tilapia Fish Cultivation, using APBDes (*Anggaran Pendapatan dan Belanja Desa*/Village Revenue and Expenditure Budget).

Bangelan Village consists of four hamlets. Each hamlet has a different focus on developing potential, managed by each farmer group. For example, in Bangelan Village, there are five farmer groups: Karya Utama 1 (Bangelan Hamlet), which controls the potential in the form of rice plants; Karya Utama 2 (Arjomulyo Hamlet), which controls the potential in the form of coffee plants; Karya Sari (Dusun Bangelan), which controls the potential in the form of goats; and Karya Mulya (Sidomulyo Hamlet), which controls the potential in the state (medium-term development plan, Bangelan Village 2019–2025).

*4.2. Agricultural Characteristics*

4.2.1. Pujon Kidul Village

The main economic sector of Pujon Kidul Village is the agriculture sector; 39.3% of the population is engaged in the agricultural sector. This percentage includes 194 people who work as farm laborers, 1334 people who work as farmers, and 288 people who work as breeders. The livelihoods of crusty rural communities in the agricultural sector, where this sector is, are supported by fertile village soil conditions. The main food crops in Pujon Kidul village are rice, fruit, and vegetable farming; based on the 2019 village profile data, the rice field area of Pujon Kidul Village has an area of 25% of the village area or 82.88 hectares (Figure 2a).

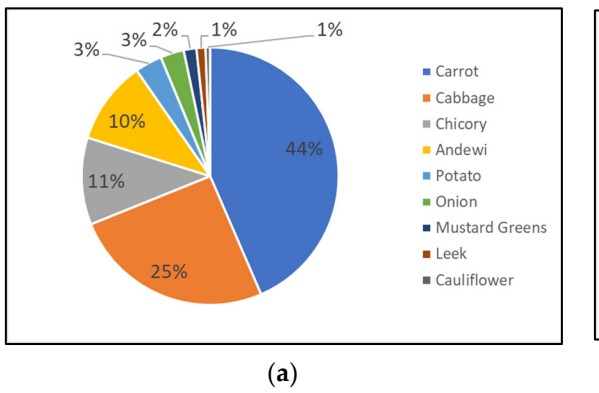
(**a**)

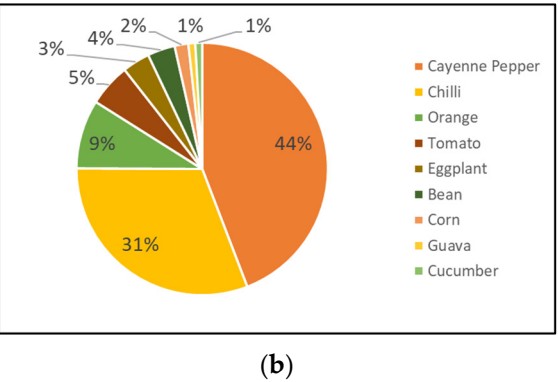
(**b**)

**Figure 2.** Area of vegetable and fruit commodity in Pujon Kidul village: (**a**) vegetable commodity; (**b**) fruits commodity. Source: SIE Pujon Kidul (2019).

SIE Pujon Kidul 2019 shows that the total land area of Pujon Kidul Village is 8.78 hectares, planted with vegetables such as potatoes, andewi, mustard greens, carrots, green onions, cauliflower, and shallots. Carrots are the largest vegetable crop, making up 44% of the total area of vegetable commodities or 3.82 hectares.

In addition to vegetables, Pujon Kidul Village is also planted with fruit commodities covering an area of 5.66 hectares. Some of the types of fruit grown are cucumbers, corn, carrots, guavas, oranges, eggplants, chilies, tomatoes, and beans. SIE Pujon Kidul 2019 shows that as much as 29% of 5.66 hectares of the fruit area is mostly planted with chili. This chili commodity consists of 1.75 hectares of large chili commodities, and 2.5 hectares of chili commodities (Figure 2b).

In the livestock sub-sector, 228 people work as farmers. Most of these farmers raise dairy cows to support their economy. The number of dairy cows in Pujon Kidul Village is 1034 heads, with each household's average number of dairy cows being 2–4 heads. Farmers typically sell their cows' milk production to cow's milk shelter cooperatives and use various processed products. Products produced by the community from processed cow's milk include yogurt, milk crackers, ice cream, milk sticks, etc.

### 4.2.2. Bangelan Village

The land area of Bangelan Village is 768.10 ha (Figure 3). There are plantations planted with coffee commodities covering most of Bangelan Village. This condition is consistent with the primary activities of rural communities, which are dominated by agriculture, plantations, and dry fields. In addition, a small portion of the plantation land is also used to grow cassava, sugar cane, toga plants, vegetables, chili, cucumbers, and tomatoes for the people of Bangelan Village.

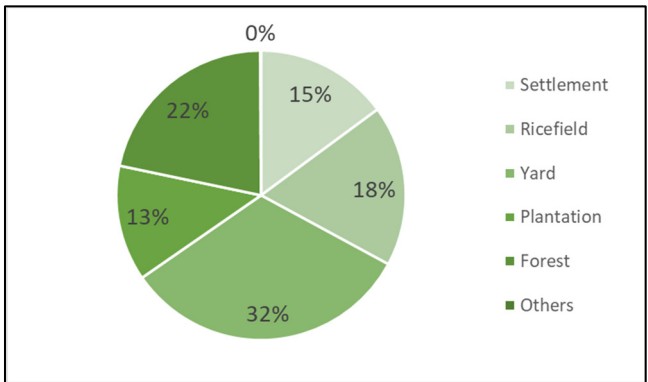

**Figure 3.** Distribution of Land Use in Bangelan Village.

The most dominant activity in Bangelan Village is the plantation sector. The land use of Bangelan Village in each of its hamlets is mostly plantation/moorland, both owned by the people of Bangelan Village and owned by PT Perseroan Terbatas/Limited Liability Company) Perkebunan Nusantara 12, or PTPN12. Rice and corn farming are also part of the agricultural activities in Bangelan Village. In addition to planting rice, rice fields are usually also interspersed with planting secondary crops, also called intercropping. The land in Bangelan Hamlet is suitable for growing rice.

### 4.3. Characteristics of Respondents

### 4.3.1. Pujon Kidul Village

The characteristics of the respondents in Pujon Kidul Village were classified by age, gender, income, education, and occupation. The age characteristics of community respondents were dominated by productive ages 15–65 years, making up a percentage of 96% or 303 respondents (Table 1). Furthermore, the level of education greatly affected the economy of the community, the majority of whom work in the agricultural sector. The characteristics of education that tended to be high were able to influence the mindset of farmers to be skilled or capable of managing agricultural products better and more efficiently. As a result, low farmer education also affects the low income below the UMK (*Upah Minimum Kabupaten*/District Minimum Wages) Malang Regency.

Based on the education received by 334 respondents, it is known that general education is very diverse, namely elementary, middle, high school, to undergraduate. Most respondents were people with the last education of elementary school, namely 59% of the total respondents or as many as 184 respondents. In addition, there were 4% of respondents who did not have formal education. Respondents' income was closely related to work (Table 1). Income can mean the receipt of money or goods. In this study, income was classified based on the Malang Regency UMK, per month. Most people had incomes below the UMK, and most of them worked as farmers in Pujon Kidul Village.

**Table 1.** Characteristic Respondents Pujon Kidul Village.

| Characteristic | Category | (%) | Characteristic | Category | (%) |
|---|---|---|---|---|---|
| Gender | Male | 74% | | Mechanic | 1% |
| | Female | 26% | | Casual daily | 2% |
| Age | 15–65 | 96% | | Farm workers | 17% |
| | >65 | 4% | | Freelancer | 1% |
| Education | Does not go to school | 1% | | Teacher | 1% |
| | Primary School | 59% | | Housewife | 1% |
| | Junior High School | 22% | | Parking attendant | 1% |
| | Senior High School | 12% | | Head of building | 1% |
| | Diploma | 4% | | Construction worker | 2% |
| | Bachelor Degree | 1% | Type of work | Trader | 10% |
| Total Income | <Rp 3,068,275 | 80% | | Civil servant | 1% |
| | Rp 3,068,275 | 15% | | Private employee | 1% |
| | | | | Village apparatus | 1% |
| | | | | Timber | 1% |
| | | | | Farmer | 42% |
| | | | | Breeder | 3% |
| | >Rp 3,068,275 | 5% | | Driver | 1% |
| | | | | Unemployment | 4% |
| | | | | Trail driver | 1% |
| | | | | Run a private enterprise | 3% |
| | | | | Entrepreneur | 2% |

### 4.3.2. Bangelan Village

The characteristics of respondents in Bangelan Village were classified by age, gender, income, education, and occupation. The characteristics of the age of community respondents were divided into age ranges of five years. Productive age is categorized in the age range of 15–65 years. In total, 97% of 312 respondents had a productive age, while those of a non-productive age, >65 years, made up 3% of respondents (Table 2).

**Table 2.** Characteristic Respondents Bangelan Village.

| Characteristic | Category | (%) | Characteristic | Category | (%) |
|---|---|---|---|---|---|
| Gender | Male | 65% | | <Rp 3,068,275 | 83% |
| | Female | 35% | Total Income | Rp 3,068,275 | 14% |
| Age | 15–65 | 97% | | >Rp 3,068,275 | 3% |
| | >65 | 3% | | Unemployment | 6% |
| Education | Does not go to school | 1% | | Farm workers | 13% |
| | Primary School | 18% | | Private employee | 2% |
| | Junior High School | 50% | | Village apparatus | 1% |
| | Senior High School | 18% | Type of work | Craftsman | 1% |
| | Bachelor Degree | 11% | | Farmer | 46% |
| | | | | Breeder | 27% |
| | >Bachelor Degree | 2% | | PTPN's day laborer | 2% |
| | | | | Trader | 2% |

The community's economy, especially those based on agriculture, is strongly influenced by educational attainment. Based on the education of 323 community respondents, it

is known that community education ranges from undergraduate, high school, junior high school, to elementary school. As much as 50% of the total, or as many as 160 respondents, had a final education in junior high school. However, in Bangelan Village, there are also as many as 1% of respondents who do not go to school.

The pie chart on (Table 2) depicts the percentage of respondents by occupation. Farming makes up a 46% share of total respondents' livelihoods. This is due to Bangelan Village's natural resources, which are ideal for agricultural activities, particularly rice and coffee production. Livelihoods influence the income of the Bangelan Village community. Most of the people of Bangelan Village make a living as farmers with uncertain production results, so the income obtained is relatively low. The table below (Table 2) shows the percentage of respondents' monthly incomes. A total of 83% of respondents have an income below the UMK Malang Regency.

## 5. Discussion

This study used SEM to develop a social capital model in dealing with a pandemic consisting of trust variables, social networks, and norms in the two research locations, Pujon Kidul Village and Bangelan Village. SEM analysis was carried out by linking social capital variables with their indicators. The following are the stages of SEM analysis in Pujon Kidul Village and Bangelan Village.

### 5.1. Social Capital in Pujon Kidul Village

Social capital analysis was conducted using SEM modeling on social capital variables in Pujon Kidul Village. In the first stage, all social capital indicators and variables were entered into the model (Figure 4a). Then, the loading factor value was calculated, where indicators that have a value of 0.5 were discarded. In this first stage, some indicators, namely K2, K3, K5, K6, J2, J3, and J4, were eliminated. Furthermore, after eliminating invalid indicators, the loading factor value was recalculated in the new model. In the second stage, the J2 indicator was removed, so that the indicators formed in the next stage included K1, K7, K4, N1, N2, N3, JI, and J3, and produced the model in the figure below (Figure 4b). The third stage of SEM testing was then repeated by recompiling the model, as shown in (Figure 4c). The results of the third stage of SEM social capital were the final stage because all indicators were valid. This social capital indicator was interpreted to describe social capital variables well in Pujon Kidul Village.

The next step was to carry out the feasibility of the SEM model using the GOF criteria in the three stages. The results of the GOF test showed a comparison of the values of stages 1, 2, and 3, based on the Chi-square value (cut of value: $<\alpha.df$ ($\alpha = 0.005$)), probability value (cut of value: $\geq 0.05$), CMIN(Chi-Square)/DF (degree of freedom) value (cut of value: $\leq 2.00$), TLI (Tucker-Lewis Index) value (cut of value: $\geq 0.90$), CFI (Comparative Fit Index) value (cut of value: $\geq 0.90$), and RMSEA (Root Mean Square Error Of Approximation) value (cut of value: $\leq 0.08$). In the first stage, the Chi-square value decreased from 203.8 to 33.0 in the second stage and to 27.0 in the third stage. The probability value increased from 0.000 to 0.103 in the second stage and to 0.057 in the third stage. The CMIN/DF value changed in the first stage to 2.343, in the second stage to 1.376, and in the third stage to 1.591. The TLI value increased by 0.785 in the first stage, 0.965 in the second stage, and 0.955 in the third stage. Likewise, the CFI value increased by 0.822 to 0.977 in the second stage and to 0.973 in the third stage. The RMSEA value decreased from 0.068 to 0.036 in the second stage and to 0.045 in the third stage.

The results of the three SEM stages showed a change in the GOF value, where the model was said to be feasible if it had four to five GOF indices that met the requirements (Haryono 2014). Stage 1 of the model could not be accepted, while stages 2 and 3 were acceptable capital. However, when compared between the three models, model 3 was a model that had all valid indicator values and had the most fit value. The third stage was the capital that could best describe social capital in Pujon Kidul Village.

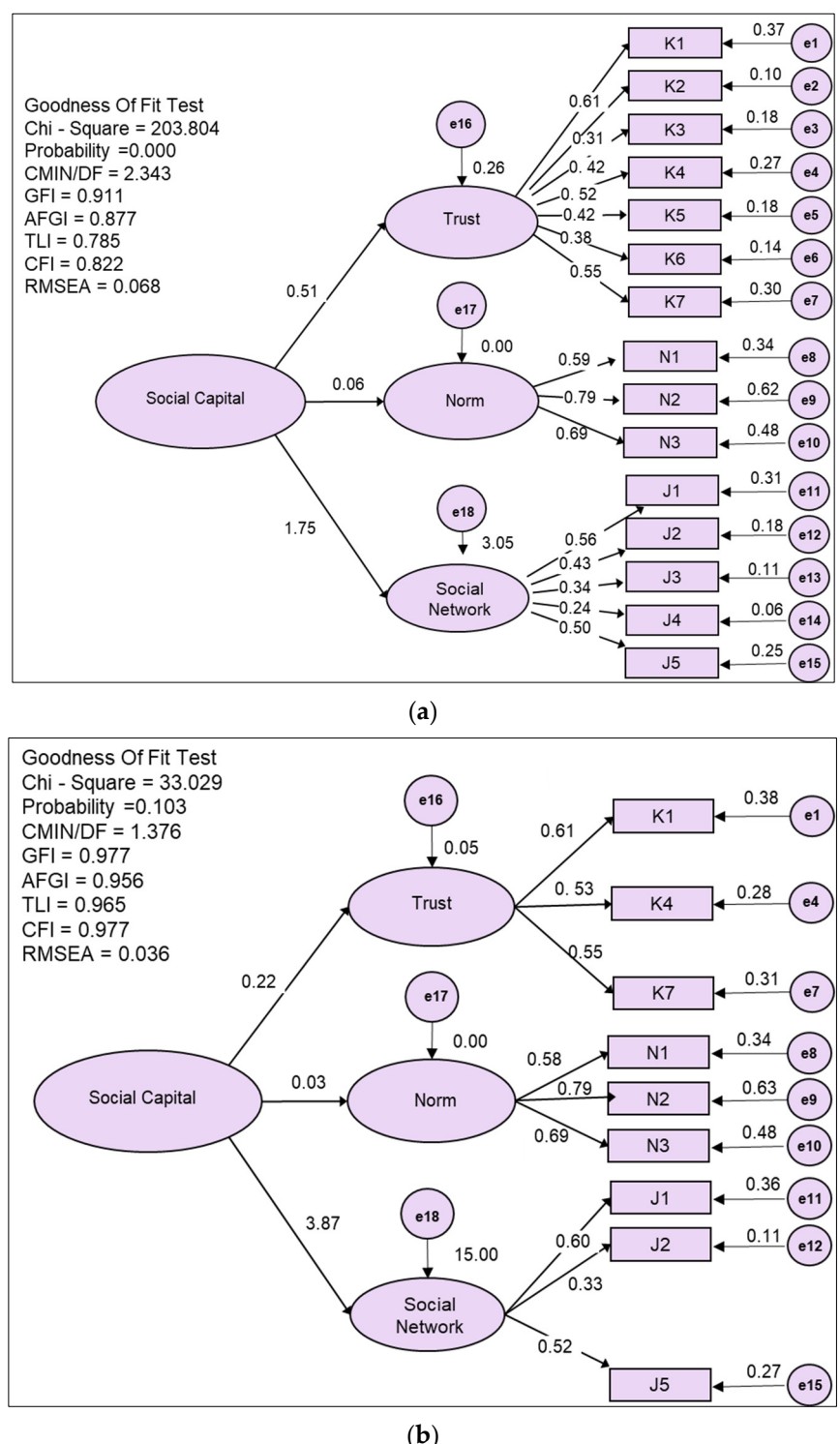

**Figure 4.** *Cont.*

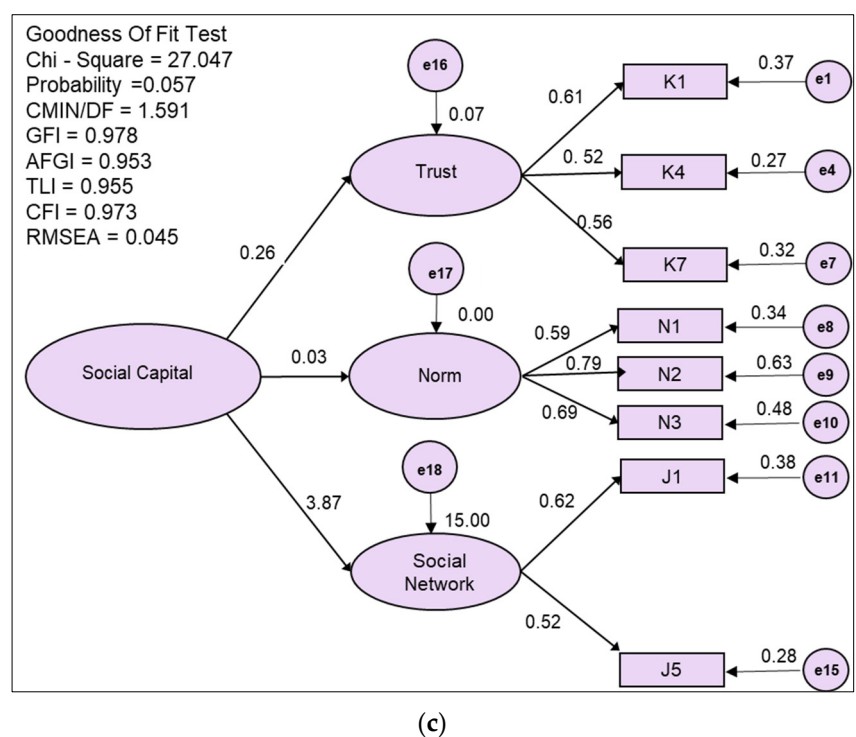

(**c**)

**Figure 4.** (**a**) Stage 1 Social Capital Pathway Diagram. (**b**) Stage 2 Social Capital Pathway Diagram. (**c**) Stage 3 Social Capital Pathway Diagram.

*5.2. Social Capital in Bangelan Village*

The Bangelan Village SEM analysis consisted of two stages to be able to describe the model that describes the social capital of the Bangelan Village community with the pandemic data. Stage 1 of the Bangelan village SEM model was carried out by entering all variables and indicators of social capital into the model and calculating the factor loading value. At this stage of analysis, indicators that had a value of 0.5 were discarded. The factors that were eliminated in stage 1 were the indicators K5, K7, J4, and J5 (Figure 5a). Then, a second test was carried out by recalculating the variables and indicators of the results of the first stage. Therefore, the second stage of the SEM test of social capital was formed from the indicators K1, K2, K3, K4, K6, N1, N2, N3, J1, J2, and J3 (Figure 5b). The results of the second stage of SEM social capital were the final stage because all indicators had met the requirements and the indicators were able to form the social capital of the village community.

Then, the next step was to carry out the feasibility of the SEM model in Bangelan Village using the GOF criteria at both stages based on Chi-square value (cut of value: $<\alpha$.df ($\alpha = 0.005$)), probability value (cut of value: $\geq 0.05$), CMIN/DF (cut of value: $\leq 3.00$), TLI value (cut of value: $\geq 0.90$), CFI value (cut of value: $\geq 0.90$), and RMSEA (Root Mean Square Error of Approximation) value (cut of value: $\leq 0.08$). The first stage of the feasibility test has a Chi-square value of 138.8 and becomes 476.03 in the second stage. Likewise, the probability value decreased from 0.052 to 0.040 in the second stage. The CMIN/DF value changed in the first stage by 1.58 to 1.365 in the second stage. The TLI value increased from 0.884 in the first stage to 0.950 in the second stage. Then, the CFI value increased by 0.904 to 0.963 in the second stage. The RMSEA value decreased from 0.047 to 0.037 in the second stage. The results of the two SEM stages showed a change in the GOF value, where the model was said to be feasible if it had four to five GOF indices that met the requirements (Haryono 2014). Thus, based on the results of the comparison of the feasibility test of the two stages, the first stage was an unfit model and the second stage was a fit model. Therefore, stage 2 of the social capital was the most fit SEM model and could describe social capital in Bangelan Village well.



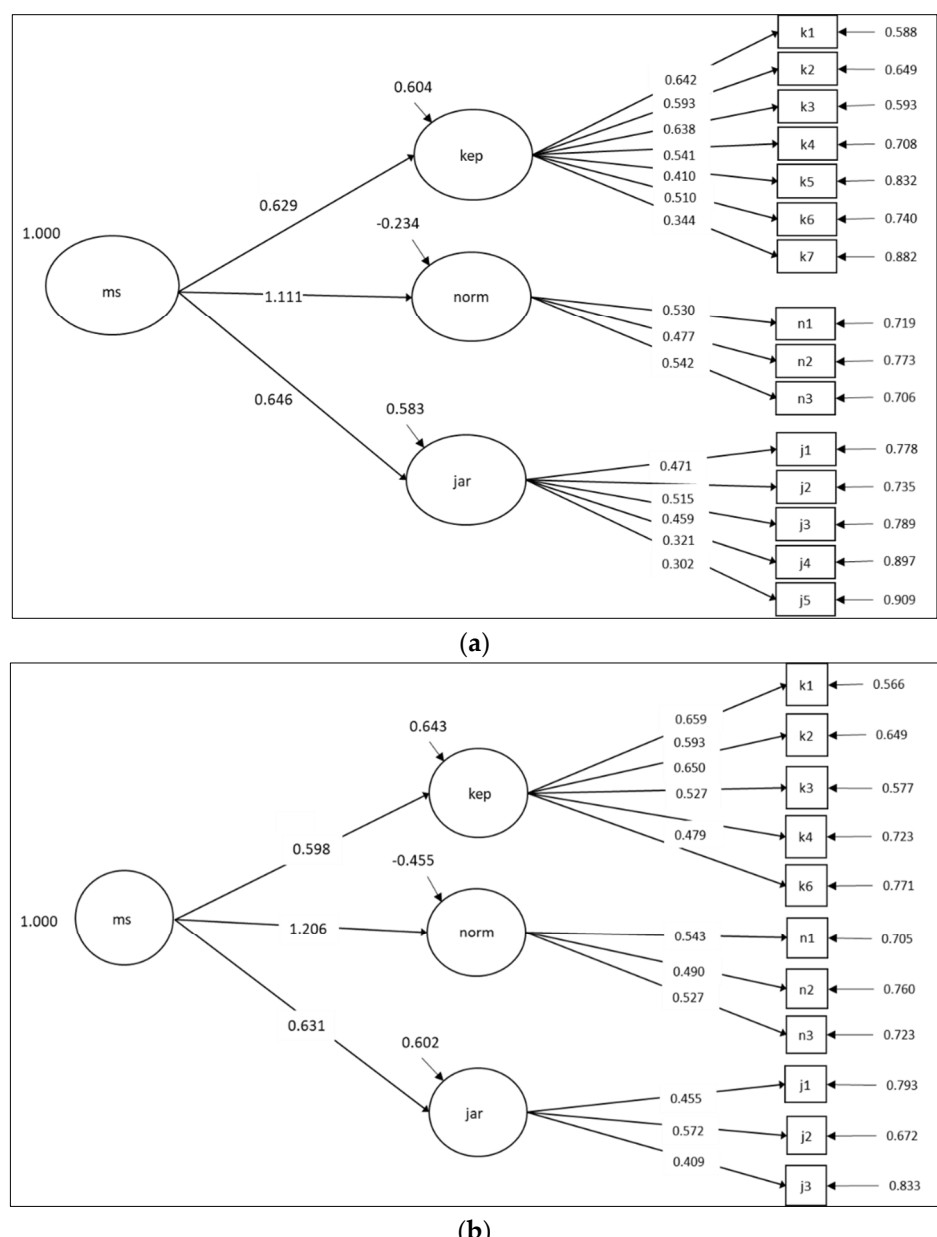

**Figure 5.** (**a**) Stage 1 Path Diagram. (**b**) Stage 2 Path Diagram.

### 5.3. Comparison of Social Capital Model Results

Social capital is a concept that leads people to develop networks, trust, and be bound by social norms. The social capital will then facilitate a mutually beneficial cooperation and coordination in achieving common goals (Lang and Hornburg 1998). Social capital refers to a network of trust, and norms that can be used to solve a common problem, especially in the current pandemic situation. There is social capital for the community in Bangelan Village and Pujon Kidul Village, which provides a good position for the government and the community to deal with pandemic problems. According to SEM analysis in both villages, the relationship between variables represented by trust, social norms, and social networks is positively influenced by social capital variables.

The formation of the social capital model in Pujon Kidul Village includes three stages, where in the first and second stages, the model only meets one GOF eligibility criteria, while in the third stage the model has fulfilled all the criteria. Therefore, in the third stage, this SEM model describes the social capital variables formed by several indicators. The dimension of confidence is described by a K1 indicator of 0.61, a K4 indicator of 0.52, and a

K7 indicator of 0.565. The K1 indicator is the indicator that has the greatest influence in shaping/describing the trust of the village community, which means that the community has high trust in their neighbors to help each other when experiencing difficulties, especially during the current pandemic crisis. A study conducted by Haj-Salem and Chebat (2014) demonstrates the existence of a high-quality relationship, which in this study is assumed to be trust between village communities, so it can withstand the negative effects of various negative events on attitudes and behavior among fellow communities. In addition, according to (Su and Swanson 2017), interpersonal relationship quality plays a crucial role in the creation and maintenance of sustainable tourism development. Despite being affected by the COVID-19 pandemic, the community in Pujon Kidul Village continues to provide information on what is required during the pandemic, including preparing a place where people can wash their hands and participate in community outreach activities related to COVID to maintain tourism development in the village, despite the affected community.

Then, the norm dimension is described by the N1 indicator of 0.585, the N2 indicator of 0.795, and the N3 indicator of 0.69. Thus, the N2 indicator is the most influential indicator in describing the norm variable, meaning that the existence of social sanctions affects the level of norms adopted by the community. The Pujon Kidul Village community applies social sanctions to people who violate norms in the form of reprimands, family settlements, and settlements with the authorities. The phrase "social norm" refers to what a group of people typically say when defining what is socially acceptable or unacceptable behavior, driven by the apprehension of social rejection and/or repercussions (Culiberg and Elgaaied-Gambier 2016). As a result, social norms are essential to social life because they guide people's actions, convictions, and emotions. Additionally, by incorporating the social norms that individuals choose to uphold, they directly contribute to the development of personal norms (Bamberg and Möser 2007). Social norms can clarify how group values affect how people make decisions for themselves. Tourism significantly affects the damage to the natural and socio-cultural environments on which it depends. The theory of value-belief and the activation of norms has been used by researchers to explain why conference attendees make pro-environmental decisions (Han 2015). The people of Pujon Kidul Village still adhere to the rules that are in place when there are tourist activities taking place. This is done to ensure that both the local populace and visitors uphold the appropriate standards of conduct and to indirectly invite visitors to do the same.

Then, the social network dimension is described by a J1 indicator of 0.62 and a J5 indicator of 0.525. Thus, the J1 indicator is the indicator that has the greatest influence, meaning that the willingness of the community to build cooperation for common goals without expecting personal gain affects the level of social network formation in Pujon Kidul Village. The network is the result of equal collaboration with different interests. The relationship between people who share the same interests and goals is described by types of networks, one of which is bonding (Pretty 2002). This relationship manifests itself in various types of groups and people who assist one another at the community level. In order to promote sustainable tourism in Pujon Kidul Village during the COVID-19 pandemic, the community there shares a common objective. During the pandemic, there was a decrease in the number of tourists at all tourist attractions, especially in Pujon Kidul Village. Communities are still swarming to implement economic recovery by relying on agricultural activities, as well as tourism-related ones, such as processing agricultural products to increase added value and income. It can be seen that the community does not depend on cattle farming, which is a superior commodity in the village of Pujon Kidul. The community is also actively involved in processing bamboo commodities. It is thought that the bamboo clump in Maron Hamlet, Pujon Kidul Village, has great potential. Apart from being a tourist attraction, the bamboo in the hamlet can be sold as souvenirs. This is also supported by Pujon Kidul Village, which has abundant resources, with over 15 ha of bamboo groves. Since entrepreneurship is thought to have the power to propel a community's economy in the direction of a more developed economy in the future, it will ultimately help the community support new entrepreneurs.

Unlike Pujon Kidul Village, Bangelan Village only requires two stages to fulfill GOF eligibility. In the second stage, the Bangelan Village SEM model describes the social capital variables formed by social capital indicators. In the dimension of confidence, the K1 indicator is 0.659, the K2 indicator is 0.593, the K3 indicator is 0.650, the K4 indicator is 0.527, and the K6 indicator is 0.527. Therefore, the K3 indicator is the most influential indicator in describing the trust variable, meaning that the Bangelan Village community has a high sense of trust in the village government because the government involves the community well in programs implemented in the village. A tourist destination is made up of a complex of numerous organizations with numerous stakeholders. It has been shown that a lack of trust among communities can indirectly hinder the development of green tourism destinations (Su et al. 2020). In order to develop visitor trust, satisfaction, and identification, a relationship's quality is also crucial for tourism (Lee et al. 2021). According to (Coombs and Holladay 2005) and (Gelbrich 2010), stakeholders' perceptions of responsibility are a crucial element in determining how the behavior of the village community is shaped. It is evident that people are working together to find a solution to a problem during the COVID-19 pandemic, one of the external crisis issues facing tourist destinations (Chi et al. 2022). In this case, the issue is an effort to keep developing Bangelan Village's tourist destinations. Due to the lack of activity of village institutions such as LPMD (*Lembaga Pemberdayaan Masyarakat Desa*/Village Community Empowerment Institutions) and BUMDEs (*Badan Usaha Milik Desa*/Village-Owned Enterprises) (they only started doing so at the end of 2020) and the low participation of teenagers or young people, the development of Bangelan Tourism Village has not been optimal thus far (Wardani et al. 2022). However, the community's ability to develop the potential of their tourism village is made easier by a high level of trust among community members and the government. These outcomes are consistent with research showing that trust significantly influences community collective action (Nugraha et al. 2021). Social capital is created through effective social management planning, which includes goals, objectives, agreements and commitments, standards, transparent practices, institutional arrangements and spaces for dialogue, and cooperation and collaboration between community actors and various levels of management, most notably the village government.

Furthermore, the norm dimension is described by a N1 indicator of 0.543, a N2 indicator of 0.490, and a N3 indicator of 0.527. Thus, the N1 indicator is the most influential indicator in describing the norm variable, meaning that adherence to customary rules and norms is the most influential indicator in describing norms in Bangelan Village. Obedience to customary norms is marked by the behavior of people who obey customary norms as a habit of living in the village. Folks who adhere to customary norms as a habit of living in the village show their compliance with customary norms through their behavior. Customary norms play an important role in changing customer intentions regarding various products and services. For example, tourists' attitudes may cause them to do something, but if it conflicts with their personal standards, they tend to do something else (Perugini and Bagozzi 2001). During the COVID-19 pandemic, the regulations that applied to tourist attractions were to maintain distance in all activities, one of which was when eating. Communities in Bangelan Village that have a place of business in the form of a restaurant in the tourist area of Bangelan Village have implemented the same rule. This is what ultimately encourages tourists to also respect and apply the rules while visiting tourist attractions in Bangelan Village.

Furthermore, the social network dimension is described by a J1 indicator of 0.455, a J2 indicator of 0.572, and a J3 indicator of 0.409. Therefore, the J2 indicator is the most influential indicator in shaping the social network, meaning the participation of the community in being involved in religious activities in Bangelan Village. These religious activities can be in the form of recitation, tahlilan, and others. A social network is defined as "a social phenomenon consisting of entities linked by special ties that reflect interaction and interdependence, such as friendship, kinship, exchange of information, etc." (Carpenter et al. 2012). The Manten coffee ritual, held prior to the harvest season and involving the grinding

of robusta coffee, the primary product of Bangelan Gardens, is one of the well-known and frequent religious practices carried out in Bangelan Village. Participants included the District Leadership Communication Forum (Forkopimka) of Kromengan District and Wonosari District, the village heads and hamlet heads of nearby Bangelan Gardens, as well as the ranks of community and religious leaders of nearby Bangelan Gardens, all of whom were present while the pandemic's health protocol was still being followed. In light of this occasion, it is hoped that all of this year's coffee picking and processing will go smoothly, and produce the highest quality and quantity possible. The continuation of this custom may help to inadvertently strengthen interpersonal friendships. Additionally, the presence of routine traditions can boost public confidence in the government and programs relating to tourism-related activities, resulting in a minimum of conflict between communities. Due to economies of scale, knowledge sharing, and marketing synergies, establishing a network of business associations and community members can benefit both parties more (Mair et al. 2016; Torres et al. 2019).

Based on the results of a comparison of the two villages, it can be seen that each village has different characteristics of social capital. In the trust variable, trust between communities in Pujon Kidul Village is higher than in Bangelan Village. However, the public's trust in the government, especially government programs related to the development of tourism activities, in Bangelan Village is higher. By bringing the community closer to routine traditions, Bangelan Village indirectly strengthens its relationship with the village government by bringing the community closer to how these traditions are carried out, in contrast to Pujon Kidul Village, where the common tradition is tahlilan. This tradition is carried out in only a few groups and does not involve all village communities directly so the information obtained on village development is not channeled perfectly. This has resulted in some of Pujon Kidul community not supporting the development of tourism in their village (because of the possibility of differences in the information received).

In the norm variable, the two villages apply the prevailing customary norms. The existence of this can improve good relations between tourists and the local community because tourists also indirectly try to respect existing norms, and they believe that adherence to norms will be considered a good thing by the local community (Cialdini et al. 1990).

In addition, in the social network variable, it can be seen that Pujon Kidul Village and Bangelan Village have different focuses on important indicators. Pujon Kidul Village applies social network variables to try to implement cooperation without prioritizing personal matters. During the COVID-19 pandemic, other commodities were processed to ensure economic recovery, so people were not overly dependent on tourism activities, and made the most of other opportunities. Even though the impact felt by the community was not direct, it could boost the income of the affected local community. The Bangelan Village community also did the same thing. The coffee commodity, which w a superior potential in the village, is processed by the community not only in the form of raw products but in the form of finished products, which are marketed at several tourist spots in Bangelan Village. In addition, the Manten Kopi ritual tradition, which is held ahead of the harvest and grinding season of robusta coffee, also has the potential to become a tourist attraction in Bangelan Village, with the aim of increasing the village revenue and expenditure budget (APBDesa) while increasing tourism development.

## 6. Conclusions

Social capital is a resource that shows up as network access, solidaristic relationships, norms, laws, and sanctions, as well as in political cooperation and participation. A community's capacity to engage in collaborative problem-solving processes is demonstrated by its social capital. This study is meant to serve as a starting point for further investigation into how collective action decisions to aid in the recovery from the pandemic crisis in the two study areas, Pujon Kidul Village and Bangelan Village, are influenced by community adaptation related to social capital. The conclusions of this study include:

1. The results of the SEM model show that the social capital model of Pujon Kidul Village is described in three stages, while the social capital model of Bangelan Village is described in two stages.
2. The social capital variable in Pujon Kidul Village is influenced by three dimensions and some of its forming indicators. The most influential indicator in describing trust is trust in neighbors. In the social network dimensions, it is most influenced by the community's willingness to cooperate, and in norms, it is most influenced by the indicators of the existence of sanctions against the norm.
3. The social capital variable in Bangelan Village is influenced by three dimensions and some of its forming indicators. The most influential factor in describing trust is trust in the village government. In social network dimensions, it is most influenced by indicators of participation in religious activities and events, and in norms, it is most influenced by indicators of adherence to customary norms and rules.

Due to the general and imprecise nature of the explanations provided, the studies conducted in this study had a limited scope, particularly those focusing on community action in implementing economic recovery during the COVID-19 pandemic. Furthermore, the analysis of this study is restricted to the village level at the local level. For definite results regarding the relationship between social capital, collective action in the COVID-19 pandemic, and further economic recovery, a comparative analysis with other post-pandemic tourist destinations must be conducted going forward (by comparing incomes before and after COVID-19). The results can then be used by public policy makers to fortify current social capital and get ready for shocks such as the COVID-19 pandemic.

On the basis of the Strategic Plan of the Ministry of Agriculture for 2020–2024 (*Keputusan Menteri Pertanian Republik Indonesia*, n.d.), the regional government can implement several policies to support sustainable agriculture in tourist villages, including:

1. Increased business partnerships, entrepreneurial training for young people, and access to finance for agricultural entrepreneurs are all part of the strategy being used to strengthen economic resilience for quality and equitable growth. Along with the expansion of agricultural industrial products, additional productivity improvements, supply chain strengthening, mechanization, product development, and product promotion are also carried out.
2. Boost infrastructure to aid in the provision of essential services and economic growth. Due to the poor performance of the operation and maintenance of irrigation systems, infrastructure support in the agricultural sector still needs to be improved, for example, in the management of water resources to support food security and nutrition. Improving the effectiveness of irrigation water allocation, building new irrigation networks, repairing existing irrigation networks, expanding irrigation institutions, and improving sub-optimal land use by reviving swamplands, are all ways to improve irrigation systems' efficiency and performance.
3. The creation of agribusiness microfinance institutions and the facilitation of agricultural insurance schemes for the revival of farmer financing.
4. Strengthening processing facilities, post-harvest handling, and marketing at the farmer/farmer group level, as well as downstream technological innovations to increase the added value of industrial-scale products, will increase high synergy with the industrial and trade sectors and boost the added value and competitiveness of agricultural products.
5. Developing innovative agricultural technologies through an integrated and sustainable research and development process in collaboration with various stakeholders (agricultural research institutions and users).
6. Enhancing the caliber of agricultural human resources through the promotion of young agrarian entrepreneurs in association with academic institutions and the private sector, as well as the dissemination of agricultural knowledge through electronic, print, and online media.

7. In order to ensure food availability and boost the competitiveness and added value of agricultural products during and after the COVID-19 pandemic, production capacity was increased, local food was diversified, food reserves were strengthened, logistic systems were strengthened, modern agriculture was developed, and the triple export movement was engaged (Gratieks).

**Author Contributions:** Conceptualization, A.A. and G.P.; methodology, A.A.; software, L.E.W.; validation, I.R.D.A., G.P. and C.M.; formal analysis, A.A. and G.P.; investigation, A.A.; resources, R.; data curation, A.A.; writing—original draft preparation, G.P. and A.A; writing—review and editing, G.P.; visualization, A.A.; supervision, I.R.D.A.; project administration, G.P.; funding acquisition, G.P. and R. All authors have read and agreed to the published version of the manuscript.

**Funding:** This research was funded by by Penelitian Thesis Magister (PTM) DRPM DIKTI with LPPM Universitas Brawijaya, Contract Number 1071.64//UN10.C10/TU/2022, and The APC was funded by DRPM DIKTI and LPPM Universitas Brawijaya.

**Institutional Review Board Statement:** Not applicable.

**Informed Consent Statement:** Not applicable.

**Data Availability Statement:** Not applicable.

**Acknowledgments:** We would like to express our great thanks to anonymous reviewers and academic editor for helpful comments on the earlier manuscript. We also thank the Journal Publishing Board (BPJ/*Badan Penerbitan Jurnal*) Faculty of Engineering, University of Brawijaya, for coaching in preparing the manuscript for the master's student.

**Conflicts of Interest:** The authors declare no conflict of interest.

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
