# Peer review of "The Role of Social Capital Facing Pandemic COVID-19 in Tourism Village to Support Sustainable Agriculture (Empirical Evidence from Two Tourism Villages in Indonesia)"

_economies, doi:10.3390/economies10120320_

Round 1
Reviewer 1 Report
Comments to Author/s
The Role of Social Capital Facing Pandemic COVID-19 in Tourism Villages to Support Sustainable Agriculture (Empirical Evidence from Two Tourism Villages in Indonesia)
In this research paper, the Author/s aimed to investigate and provide knowledge about community adaptation and social capital as factors for collective action concerning economic recovery efforts following the COVID-19 pandemic. The topic is current and is worthy of a research paper. However, this study has some concerns, and some sections need improvement. I think a few issues need careful reconsideration, especially the paper’s readability, manuscript structure, and scientific soundness.
My key concern in this paper is the data analysis and research problem. Some experiments are needed for further development and are explained in the manuscript. The data analysis is not appropriate; it should be correctly done and presented scientifically. Also, there is no substantiation of the economic contribution to the problem of economic recovery effects following the pandemic. Therefore, I recommend this research paper reconsider for publication after major corrections.
In response to this article, I suggest the following:
1. The topic should be short and focus on the study scope.
2. Author/s may further improve the Abstract by incorporating the study’s key findings. Please add two or three more keywords.
3. Introduction section: This research paper mainly focuses on the social capital concept as a factor for collective action concerning economic recovery.
a. The Introduction should present the research gap, the novelty, and the study's contribution.
b. The Authors should further discuss the social capital concept and its mechanism/channels to economic recovery during the pandemic in the third paragraph of the introduction section and critically link it to the COVID pandemic. I suggest that the Authors include more relevant scientific/classic references.
4. The article does not indicate the tested hypotheses. Hence, the Authors should include a separate section for a Literature review following the introduction section. The literature section should be more specific and discuss the latest arguments and findings. The Author/s may explain the theoretical mechanism of social capital on collective action and economic recovery and discuss the key channels of social capital in detail in the introduction section as explained. Then the Author/s may develop the conceptual model with the research hypothesis.
5. Materials and Methods: The methods should be adequately described, with a suitable justification for why those methods were selected. Carefully discuss the sampling techniques. Also, I see no reason to include Table 1 in this section; it is unnecessary. Make a clear justification for the data analysis techniques employed in the study.
The Author/s has taken the social capital variable as the latent variable with three dimensions, and indicators are used in the questionnaire to measure latent variables. Please describe the indicators from past literature without bias. It is certainly helpful if the Author/s wants to ensure direct comparability of the results to previous work. This is a critical issue in this study, so I strongly request to justify your indicators using past empirical findings.
6. Results section. This discussion section should be built on the context of dialogue with researchers in the literature review. The study area can be briefly discussed in the methodology section but not in this section; also, please provide a clear discussion.
The result section is very short, only a paragraph, and it explains the indicators of some variables. I suggest the Author/s re-write this section. I hope the Author/s find a scientific system to present the characteristics of tourism villages and agricultural characteristics.
Kindly name section three as the Results and Discussion, merging sections 3.3 and 4 discussion section. In the first part of this proposed section, the authors can present the characteristics of respondents. The Author/s are advised not to submit basic information (e.g., gender, age groups) in separate charts, as it is unnecessary in a scientific paper. Table 2 needs modifications.
7. Results must be clear (to the reader) and more focused on the research objective. Can the Author/s re-write the results section of the paper? How is this paper be beneficial in terms of collective action and economic recovery? Please include suggestions and limitations.
Finally, the Author/s are encouraged to correct the academic writing mistakes in the paper. The document layout needs improvement, and proofreading is necessary.
All the best!

Reviewer 2 Report
The paper is interesting and atual.
Suggestions:
1) The theorethical framework should be improved. (Authors have integrated the theoretical discussion in the introduction).
2) The role of social capital facing pandemic should be better discussed, namely its role to support sustainable agriculture.
3) The objectives of the study could be better explained and articulated with the conclusions.
4) The title of this paper put in evidence the importance of sustainable agriculture in rural villages. However, this concept is not analysed during the paper. Authors should clarify the concept.
5) Regarding figures 6 and 7, they should follow the same format. For instance Figure 6 (a),(b),(c), (d) vs Figure 7a , figure 7b
6) Indicators should have a legend. K1?, K2? ..... (Authors mentioned the indicators in lines 71-79. However, a legend could improve the understanding of the paper results).
7) Authors should improve the conclusions. Simultaneously, they should add the study limits and future/implications.
Reviewer 3 Report
Thanks for giving the opportunity to review the paper titled as The role of social capital facing pandemic covid-19 in tourism village to support sustainable agriculture (Empirical evidence from two tourism villages in Indonesia). A This paper is timely, well developed, conducted and well written. It addresses a significant topic likely to be of interest to agriculture village destination sustainable development in rural areas. Thus, it addresses a significant topic likely to be of interest to Economies. Despite all of this, there are several possible revisions as the following:
First, it’s maybe right that author(s) point(s) out many this can positively affect sustainable tourism development for agriculture village destination. However, destination crisis event has negative impacts on tourist’s emotion, attitudes and behavior in turn influence sustainable tourism development (Su, Jia, & Huang, 2022; Su, Pan, & Huang, 2023), especially in the COVID-19 pandemic context. Please author(s) refer(s) these references, and explain why they do not consider these factors. At least, author(s) need mention them, and list these as a limitation.
“How do destination negative events trigger tourists’ perceived betrayal and boycott? The moderating role of relationship quality”. Tourism Management, 92, 104536.
“How does destination crisis event type impact tourist emotion and forgiveness? The moderating role of destination crisis history”. Tourism Management, 94, 104636.
Second, it needs point out the research gaps more clearly in the introduction section.
Third, the theoretical model of SEM needs more clearly. Specially, what’s the role of social network, norms, and trust, and the relationships among them.
Fourth, overall, the contribution of the paper and the discussion of the contribution should be made more clearly.
Fifth, it need point out the limitations and future research directions more specifically in the end of the paper.
Though the above-mentioned possible shortages, this paper is a high-quality original article. Thus, I recommend author(s) revise(s) the paper according to reviewer’s comments/suggestions.
Round 2
Reviewer 1 Report
Comments to Authors
When I carefully checked the Authors' point-to-point replies to my suggestions, I noticed that the Authors had addressed my recommendations in my review.
Further comments to Authors
1. Author/s may further improve the Abstract by incorporating the study's objectives.
2. Authors are advised not to present basic information (e.g., gender, age groups) in separate figures (For instance: Figure number 2. a, 2. b, 4 a, 5a, 5b, etc.), as it is unnecessary in a scientific paper. I suggest authors use a statistical table to present basic information in the manuscript.
3. I highly encourage authors to include more specific policy recommendations
4. Proofreading is necessary.
I recommend this research paper for publication with minor corrections.
All the best!

Reviewer 3 Report
I am satisfied with the revising works by the author(s), and recommend to accept it.
Round 3
Reviewer 1 Report
All the best..!